# Nitrogen-fixing trees could exacerbate climate change under elevated nitrogen deposition

Sian Kou-Giesbrecht[1] & Duncan Menge [1]

Biological nitrogen fixation can fuel $CO_2$ sequestration by forests but can also stimulate soil emissions of nitrous oxide ($N_2O$), a potent greenhouse gas. Here we use a theoretical model to suggest that symbiotic nitrogen-fixing trees could either mitigate ($CO_2$ sequestration outweighs soil $N_2O$ emissions) or exacerbate (vice versa) climate change relative to non-fixing trees, depending on their nitrogen fixation strategy (the degree to which they regulate nitrogen fixation to balance nitrogen supply and demand) and on nitrogen deposition. The model posits that nitrogen-fixing trees could exacerbate climate change globally relative to non-fixing trees by the radiative equivalent of 0.77 Pg C yr$^{-1}$ under nitrogen deposition rates projected for 2030. This value is highly uncertain, but its magnitude suggests that this subject requires further study and that improving the representation of biological nitrogen fixation in climate models could substantially decrease estimates of the extent to which forests will mitigate climate change.

[1] Department of Ecology, Evolution and Environmental Biology, Columbia University, New York, NY 10027, USA. Correspondence and requests for materials should be addressed to S.K.-G. (email: sk4220@columbia.edu)

Forests currently sequester a quarter of annual anthropogenic $CO_2$ emissions[1,2]. Nitrogen-fixing tree symbioses, in which bacteria living in root nodules convert atmospheric $N_2$ gas to a plant-available form of nitrogen (N), can provide much of the N needed to drive forest growth[3,4]. N-fixing trees thus mitigate climate change by sequestering $CO_2$, either directly via their own growth or indirectly via the turnover of their N-rich tissues whose decomposition increases surrounding soil N and plant growth. However, in addition to driving $CO_2$ sequestration, elevated soil N driven by the decomposition of N-rich plant litter can also drive soil emissions of nitrous oxide ($N_2O$)[5–10], a potent greenhouse gas[11]. What is the current balance of the $CO_2$ and $N_2O$ effects of N-fixing trees, i.e. the net $CO_2$–$N_2O$ effect, and to what degree will it be modified by global change?

Studies on another major N input to forests, atmospheric N deposition, offer insight into the net $CO_2$–$N_2O$ effect of N enrichment. N deposition rates are increasing globally due to fossil fuel and fertilizer use[12]. Although intensifying N deposition is expected to stimulate $CO_2$ sequestration[13], it is also expected to stimulate soil $N_2O$ emissions[14–17] that will offset 18–90% of the negative radiative forcing of this $CO_2$ sequestration[15]. These studies demonstrate the potential for elevated soil $N_2O$ emissions to substantially offset $CO_2$ sequestration driven by N enrichment. However, the balance of the $CO_2$ and $N_2O$ effects of biological N fixation, which has fundamentally different dynamics than those of N deposition, is unresolved.

Unlike N deposition, biological N fixation has the capacity to self-regulate, feeding back to ecosystem-scale soil N levels[18]. A deficiency of N can stimulate N fixation, which can promote plant growth and $CO_2$ sequestration. An excess of N can inhibit N fixation, which is physiologically costly, reducing ecosystem-scale soil N excess and its associated soil $N_2O$ emissions. However, the strength of this feedback varies across N-fixing species. Some N-fixing species exhibit a facultative N fixation strategy and feedback to soil N levels[3,18–20], downregulating N fixation rates from over 30 to 0 kg N ha$^{-1}$ yr$^{-1}$ at the ecosystem scale[3]. However, other N-fixing species do not regulate their N fixation rate in response to soil N levels, exhibiting an obligate N fixation strategy[18,21,22]. In this case, N fixation at the ecosystem scale is only downregulated when these species are competitively excluded. However, before competitive exclusion occurs, obligate N-fixing trees can drive substantial soil $N_2O$ emissions[5]. The strong connection between N fixation, soil N enrichment, and soil $N_2O$ emissions calls for the explicit consideration of N fixation strategies when estimating the net $CO_2$–$N_2O$ effect of forests.

Here we use a theoretical modeling approach to ask two main questions: how do N-fixing trees influence the net $CO_2$–$N_2O$ effect of forests, i.e. do N-fixing trees mitigate or exacerbate climate change? How will their influence change under elevated N deposition rates? We use the terms mitigate and exacerbate to highlight that the influence of N-fixing trees is relative to ongoing greenhouse gas effects. In forests, the cooling effect of $CO_2$ sequestration is partially offset by the warming effect of soil $N_2O$ emissions[2], resulting in a net cooling $CO_2$–$N_2O$ effect. We are not suggesting that N-fixing trees can or will change the direction of the net $CO_2$–$N_2O$ effect of forests from cooling to warming. The question we address is how N-fixing trees modify $CO_2$ sequestration in comparison to how they modify soil $N_2O$ emissions relative to non-fixing trees.

We use a differential equation ecosystem model that captures the fluxes and pools of carbon (C) and N in an ecosystem, and includes competition between N-fixing and non-fixing trees. We validated the model against literature estimates of the relevant fluxes and pools of C and N in tropical, temperate, and boreal forests. The model predicts $CO_2$ sequestration ($CO_2$ effect) and soil $N_2O$ emissions ($N_2O$ effect) of an ecosystem with a given

dominant N fixation strategy. We compute the net $CO_2$–$N_2O$ effect of the ecosystem with two complementary methods. The first method compares accumulated $CO_2$ sequestration to accumulated soil $N_2O$ emissions after 100 years of ecosystem succession using the global warming potential of $N_2O$. The second method computes the net radiative forcing from continuous $CO_2$ sequestration and soil $N_2O$ emissions over 100 years of ecosystem succession. To evaluate the $CO_2$ and $N_2O$ effects of N-fixing trees, we compare model ecosystems of non-fixing trees to model ecosystems that contain both N-fixing trees and non-fixing trees. Model ecosystems with N-fixing trees contain one of three empirically supported N fixation strategies[18]: obligate (fix N at a constant rate per unit biomass), perfectly facultative (downregulate N fixation to perfectly meet their N demand after taking up soil N; hereafter facultative), and incomplete regulator (downregulate N fixation similarly to the facultative strategy but sustain N fixation at a constant minimum rate). The difference in the net $CO_2$–$N_2O$ effect between a model ecosystem of non-fixing trees and a model ecosystem with N-fixing trees is the net $CO_2$–$N_2O$ effect attributed to the N-fixing trees and is inherently relative to the net $CO_2$–$N_2O$ effect of non-fixing trees. To estimate the magnitude of the net $CO_2$–$N_2O$ effect of N-fixing trees at the global scale, we parameterized the model for tropical, temperate, and boreal forests, and simulated the model under past (low; pre-Anthropocene[23]), recent (intermediate; 2001[24] and 2006[12]), and future N deposition rates (high; 2030 for the SRES A2 scenario[12,25]). The model suggests that N-fixing trees can either mitigate or exacerbate climate change relative to non-fixing trees, contingent on their N fixation strategy and on N deposition. As N deposition intensifies, N-fixing trees stimulate substantial soil $N_2O$ emissions but promote minimal $CO_2$ sequestration, exacerbating climate change relative to non-fixing trees. The goal of this study is not to generate a quantitatively accurate prediction of the net $CO_2$–$N_2O$ effect of N-fixing trees. Rather, the objectives are to estimate its potential magnitude, and to generate and explore hypotheses of how N-fixing trees could mitigate or exacerbate climate change. Ultimately, these hypotheses should be analyzed empirically and with Earth System Models.

## Results

**Net $CO_2$–$N_2O$ effect of N-fixing trees at the ecosystem scale.** Our model suggests that N-fixing trees can either mitigate climate change relative to non-fixing trees (a negative net $CO_2$–$N_2O$ effect of N-fixing trees relative to non-fixing trees) or exacerbate climate change relative to non-fixing trees (a positive net $CO_2$–$N_2O$ effect of N-fixing trees relative to non-fixing trees). The main controls that determine this balance are N fixation strategy and N deposition rate (Fig. 1 displays results for tropical forests and Supplementary Figures 1 and 2 display results for temperate and boreal forests respectively; because patterns are analogous between tropical, temperate, and boreal forests we hereafter focus on tropical forests). For N-fixing trees that exacerbate climate change relative to non-fixing trees, soil $N_2O$ emissions do not necessarily offset the absolute level of $CO_2$ sequestration (see Supplementary Figure 3 for the absolute net $CO_2$–$N_2O$ effects of ecosystems with and without N-fixing trees). Rather, the offset of $CO_2$ sequestration by soil $N_2O$ emissions for ecosystems with N-fixing trees is greater than the offset of $CO_2$ sequestration by soil $N_2O$ emissions for ecosystems without N-fixing trees. Similarly, for N-fixing trees that mitigate climate change relative to non-fixing trees, the offset of $CO_2$ sequestration by soil $N_2O$ emissions for ecosystems with N-fixing trees is lower than the offset of $CO_2$ sequestration by soil $N_2O$ emissions for ecosystems without N-fixing trees. Generally, under low N deposition rates, N-fixing trees promote $CO_2$ sequestration but

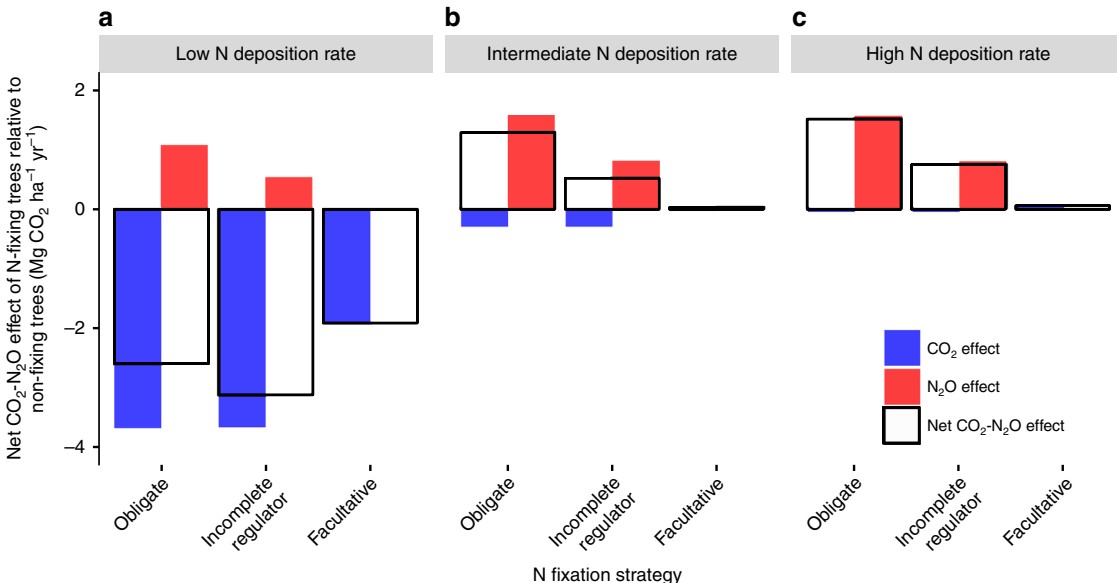

**Fig. 1** Modeled $CO_2$ and $N_2O$ effects of nitrogen-fixing trees relative to non-fixing trees. The $CO_2$ and $N_2O$ effects of N-fixing trees relative to non-fixing trees are shown under **a** low N deposition rates[23], **b** intermediate N deposition rates[12], and **c** high N deposition rates[12]. Units are $CO_2$ radiative equivalents, which balance the greenhouse effects of $CO_2$ and $N_2O$ using the global warming potential of $N_2O$. A positive net $CO_2$–$N_2O$ effect of N-fixing trees relative to non-fixing trees indicates that N-fixing trees have a warming effect relative to non-fixing trees (i.e. N-fixing trees warm more than non-fixing trees but do not necessarily warm overall). A negative net $CO_2$–$N_2O$ effect of N-fixing trees relative to non-fixing trees indicates that N-fixing trees have a cooling effect relative to non-fixing trees (i.e. N-fixing trees cool more than non-fixing trees but do not necessarily cool overall). The model is parameterized for a tropical forest

only minimal soil $N_2O$ emissions relative to non-fixing trees (Fig. 1a), whereas under high N deposition rates, N-fixing trees stimulate soil $N_2O$ emissions but only minimal $CO_2$ sequestration relative to non-fixing trees (Fig. 1c).

Obligate and incomplete regulator N-fixers sustain N fixation after satisfying their N demand, whereas facultative N-fixers shut off N fixation after satisfying their N demand (Fig. 2a). Over succession, obligate and incomplete regulator N-fixers promote greater N supply to the ecosystem via sustained N fixation than facultative N-fixers (indicated by the vertical lines in Fig. 2b, c). Under low N deposition, N supplied via N fixation by obligate and incomplete regulator N-fixing trees facilitates non-fixing trees in meeting their N demand, amplifying ecosystem-scale $CO_2$ sequestration to a greater extent than that by facultative N-fixing trees (Fig. 2b). However, this N supplied via N fixation also stimulates soil $N_2O$ emissions (Fig. 2c). This is especially pronounced for obligate N-fixers, which sustain N fixation at a higher rate than incomplete regulator N-fixers. As such, under low N deposition rates, incomplete regulator N-fixing trees exhibit the greatest net $CO_2$–$N_2O$ cooling effect because of their high $CO_2$ effect (Fig. 1a). They are followed by obligate N-fixing trees, which have a similarly high $CO_2$ effect but a higher $N_2O$ effect (Fig. 1a). Facultative N-fixing trees, which have a substantially lower $CO_2$ effect, have the lowest net $CO_2$–$N_2O$ cooling effect (Fig. 1a).

Increased N supply to the ecosystem via elevated N deposition induces N-fixing trees to downregulate N fixation to the greatest extent possible (Fig. 2a): facultative N-fixers completely downregulate N fixation and incomplete regulator N-fixers partially downregulate N fixation, whereas obligate N-fixers do not downregulate N fixation. Because facultative N-fixing trees completely downregulate N fixation (Fig. 2a), they have a negligible net $CO_2$–$N_2O$ effect relative to non-fixing trees under high N deposition rates (Fig. 1c). Under high N deposition rates, N demand is satisfied by N deposition. As such, N fixed by obligate and incomplete regulator N-fixing trees due to sustained N fixation does not contribute to $CO_2$ sequestration (Fig. 2b).

Rather, it contributes to soil $N_2O$ emissions, which increase indefinitely with increasing N fixation (Fig. 2c). Thus, obligate and incomplete regulator N-fixing trees exhibit a considerable $N_2O$ effect, yielding a net $CO_2$–$N_2O$ warming effect relative to non-fixing trees (Fig. 1c).

Initial soil N pool sizes do not strongly influence the net $CO_2$–$N_2O$ effect of N-fixing trees relative to non-fixing trees (differ by <1 Mg $CO_2$ ha$^{-1}$ yr$^{-1}$ between low and high initial soil N pool sizes; Supplementary Figure 4).

**Net $CO_2$–$N_2O$ effect of N-fixing trees at the global scale**. To ascertain how important the climate impacts of N-fixing trees could be, we estimated the net $CO_2$–$N_2O$ effect of N-fixing trees at the global scale. Although N-fixing trees play a crucial role in forests, the global distribution of N fixation strategies is not well established[26]. Accordingly, we made estimates of the global net $CO_2$–$N_2O$ effect of N-fixing trees first by examining three basic scenarios: all N-fixing trees are obligate, all N-fixing trees are facultative, and all N-fixing trees are incomplete regulators. Because forests around the globe include an assemblage of these three N fixation strategies[18,27], the maximum and minimum of these three basic scenarios provide bounds to the global net $CO_2$–$N_2O$ effect of N-fixing trees. We ran each basic scenario under future N deposition rates (for the SRES A2 scenario). Our model suggests that if all N-fixing trees are facultative, they will have an insignificant influence on estimates of the net $CO_2$–$N_2O$ effect of global forests (Table 1). At the opposite extreme, if all N-fixing trees are obligate, N-fixing trees will decrease estimates of the net $CO_2$–$N_2O$ effect of global forests by the radiative equivalent of 0.77 Pg C yr$^{-1}$ (Table 1).

In a further analysis, we determined the global net $CO_2$–$N_2O$ effects of N-fixing trees relative to non-fixing trees for a range of relative abundances of ecosystems containing obligate N-fixing trees and ecosystems containing facultative N-fixing trees under a range of N deposition rates (Fig. 3a). Under recent N deposition rates, our assumptions of the relative abundances of ecosystems

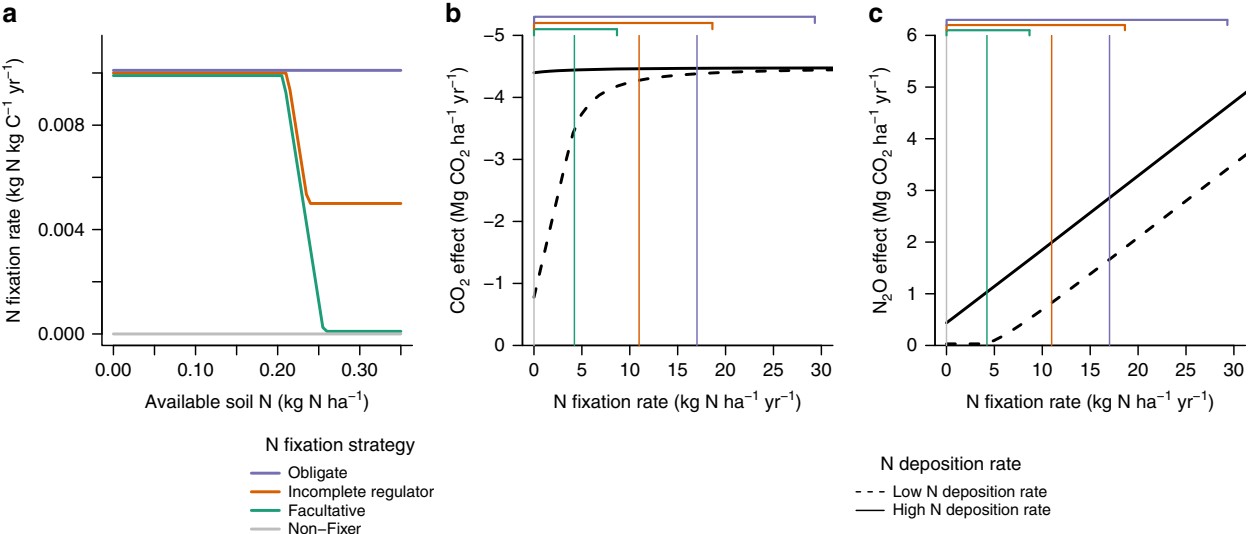

**Fig. 2** Mechanisms that drive the $CO_2$ and $N_2O$ effects of nitrogen-fixing trees. **a** N fixation rate as a function of available soil N for the three N fixation strategies examined in the model (the horizontal gray line represents a zero N fixation rate for non-fixing trees). **b** $CO_2$ effect. $CO_2$ sequestration increases with increasing N fixation rate when N is limiting. When N supply to the ecosystem is sufficient to alleviate N limitation, $CO_2$ sequestration plateaus with increasing N fixation rate. This plateau occurs at a lower N fixation rate under high N deposition than under low N deposition. **c** $N_2O$ effect (displayed in units of $CO_2$ radiative equivalents). Increasing N fixation rate does not stimulate soil $N_2O$ emissions when N is limiting. When N supply to the ecosystem is sufficient to alleviate N limitation, soil $N_2O$ emissions increase with increasing N fixation rate. This increase occurs at a higher N fixation rate under low N deposition than under high N deposition. The black curves in **b** and **c** represent the $CO_2$ and $N_2O$ effects respectively of an ecosystem with a tropical forest parameterization, a single biomass C pool, and a prescribed constant N fixation rate per unit biomass C. The vertical purple, orange, and green lines in **b** and **c** represent average N fixation rates over 100 years for the three N fixation strategies examined in the model (the vertical gray line represents a zero N fixation rate over 100 years for non-fixing trees). The corresponding brackets indicate the range of N fixation rates over 100 years for the three N fixation strategies examined in the model. The low N deposition rate is from Galloway et al.[23] and the high N deposition rate is derived from Dentener et al.[12] Overall, **a**–**c** show that N fixation drives cooling when N is limiting (low N fixation and/or N deposition) and warming when N is not limiting (high N fixation and/or N deposition)

**Table 1 Modeled global net $CO_2$–$N_2O$ effect of forests and of N-fixing trees relative to non-fixing trees under future N deposition rates (2030 for the SRES A2 scenario)**

| Global forest composition | Global net $CO_2$–$N_2O$ effect of forests (Pg C $yr^{-1}$) | Global net $CO_2$–$N_2O$ effect of N-fixing trees relative to non-fixing trees (Pg C $yr^{-1}$) |
| --- | --- | --- |
| Obligate N-fixer and non-fixer | −2.98 | +0.77 |
| Facultative N-fixer and non-fixer | −3.72 | +0.03 |
| Incomplete regulator N-fixer and non-fixer | −3.40 | +0.36 |
| Non-fixer | −3.76 | Not applicable |

Scenarios displayed are: all N-fixing trees are obligate, all N-fixing trees are facultative, and all N-fixing trees are incomplete regulators. Units are C radiative equivalents, which balance the greenhouse effects of $CO_2$ and $N_2O$ using the global warming potential of $N_2O$. Negative values in the centre column indicate a net cooling $CO_2$–$N_2O$ effect of forests. Positive values in the right-hand column, which are the differences from the non-fixer row in the centre column, indicate that N-fixing trees have a net warming $CO_2$–$N_2O$ effect relative to non-fixing trees
*NA* not applicable

containing obligate and facultative N-fixing trees have a negligible influence on the global net $CO_2$–$N_2O$ effect of N-fixing trees relative to non-fixing trees (Fig. 3b, Supplementary Table 1), whereas under future N deposition rates these assumptions can change this global scale estimate by up to 0.77 Pg C $yr^{-1}$ (Fig. 3b, Table 1).

## Discussion

Our model identifies N fixation strategy and N deposition rate as the main controls of the net $CO_2$–$N_2O$ effect of N-fixing trees at both the ecosystem and global scales (Figs. 1 and 3). In particular, under elevated N deposition rates, our model suggests that N fixation strategy is the key determinant of the net $CO_2$–$N_2O$ effect of forests: obligate N-fixing trees exacerbate climate change relative to non-fixing trees, whereas facultative N-fixing trees influence climate change in the same manner as non-fixing trees.

The net $CO_2$–$N_2O$ effect of N-fixing trees at the global scale under future N deposition rates—up to 0.77 Pg C $yr^{-1}$ according to our model—is highly uncertain, given the numerous caveats associated with scaling a simple model up to the globe. However, the magnitude of this estimate suggests that N-fixing trees could have a critical influence on the extent to which forests will mitigate climate change. Below, we discuss our current understanding of N fixation strategies and the $CO_2$ and $N_2O$ effects of N-fixing trees, how other global change factors could influence the net $CO_2$–$N_2O$ effect of N-fixing trees, and extensions of our results to forest management and Earth System Models.

According to our model, N fixation strategies are a key determinant of how N-fixing trees will influence climate change, but the global distribution of N fixation strategies is not well established. There is observational evidence that actinorhizal N-fixing trees in temperate forests are obligate[21,22] but that rhizobial N-fixing trees in tropical forests downregulate N fixation (either with a facultative

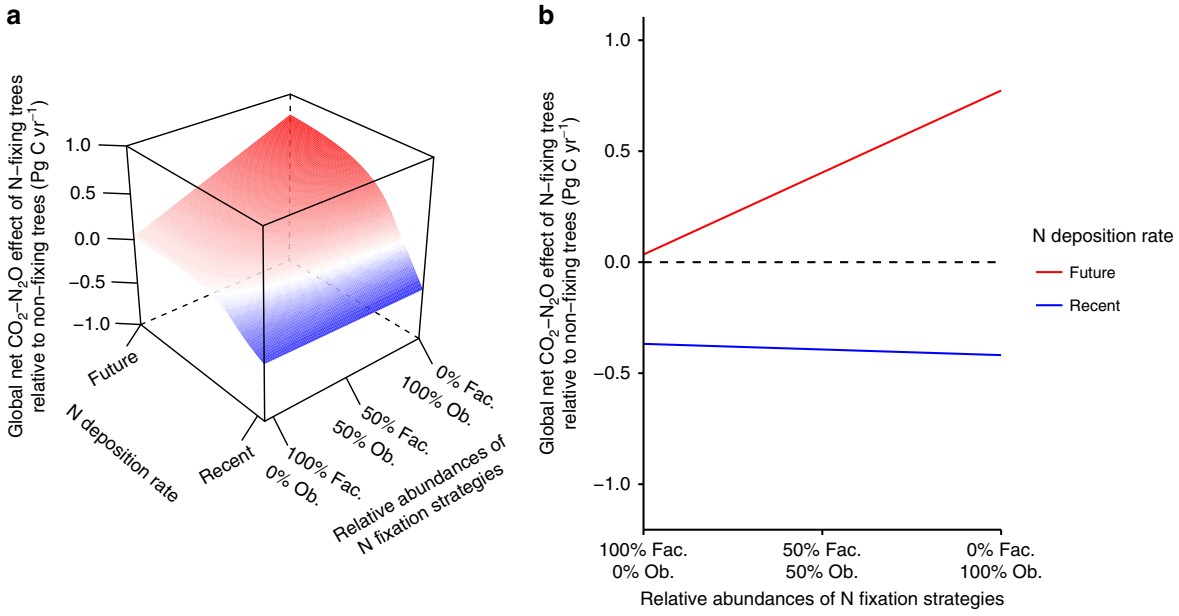

**Fig. 3** Modeled global $CO_2$ and $N_2O$ effects of nitrogen-fixing trees relative to non-fixing trees. **a** Global net $CO_2$–$N_2O$ effect of N-fixing trees relative to non-fixing trees for a range of relative abundances of ecosystems containing obligate N-fixing trees and ecosystems containing facultative N-fixing trees under a range of N deposition rates. Global forest composition ranges from the scenario in which all N-fixing trees are facultative to the scenario in which all N-fixing trees are obligate, i.e. the relative abundances of ecosystems containing obligate N-fixing trees and ecosystems containing facultative N-fixing trees range from 0 to 100% and 100 to 0% respectively. Red represents a warming effect and blue represents a cooling effect relative to non-fixing trees. Global N deposition rate ranges from the minimum recent N deposition rate derived from Vet et al.[24] or Dentener et al.[12], to the future N deposition rate derived from Dentener et al.[12]. **b** Global net $CO_2$–$N_2O$ effect of N-fixing trees relative to non-fixing trees under recent and future N deposition rates. The curves in **b** are cross sections of the extremes of the surface displayed in **a**. The dotted line is at zero, representing the transition between a cooling effect and a warming effect relative to non-fixing trees. Fac. represents ecosystems containing facultative N-fixing trees and Ob. represents ecosystems containing obligate N-fixing trees

or an incomplete regulator N fixation strategy)[3,19,20]. Theoretical evidence suggests that a transition from facultative N fixation strategies at lower latitudes to obligate N fixation strategies at higher latitudes could explain the order of magnitude drop in N-fixing tree abundance[27] and the differences in successional patterns of N-fixing tree abundance between tropical and temperate forests[28,29]. Theory also suggests why an obligate N fixation strategy could be more adaptive at higher latitudes: low decomposition rates at low temperatures could lead to sustained N limitation, favoring obligate N fixation[30]. However, there is limited empirical evidence to support these theories because N fixation strategies are difficult to establish experimentally[18]. Our study emphasizes the need for a more accurate and extensive description of the distributions of different N fixation strategies given their significant influence on predictions of the net $CO_2$–$N_2O$ effect of global forests.

The $CO_2$ sequestration component of our model relies on the theory that N-fixing trees drive forest growth by meeting its N demand, which has some[3,4] but not universal[31–33] support. For example, Batterman et al.[3] found that in a 300-year forest chronosequence in Panama, N-fixing trees provided over 50% of the N demand of early successional forest growth. However, another study from the same region of Panama showed a negligible influence of N-fixing trees on forest growth[32]. Furthermore, recent studies in Alaska[31] and Costa Rica[33] have shown that N-fixing trees can even inhibit the growth of surrounding trees and thus inhibit forest growth. These results could be due to non-N limitation and strong competitive effects of N-fixing trees on surrounding trees, although these mechanisms remain speculative. Further research is necessary to determine the predominance and controls of non-facilitative effects of N-fixing trees on forest growth. Additional studies on how N-fixing trees

drive soil $N_2O$ emissions are also necessary. It is well established that soil N drives soil $N_2O$ emissions[34,35]. However, studies of the extent to which N-fixing trees enrich soil N and stimulate soil $N_2O$ emissions are rare, although they demonstrate that N-fixing trees can substantially increase soil $N_2O$ emissions[5–10] (soil $N_2O$ emissions can be up to 12-fold greater in stands of N-fixing trees than in stands of non-fixing trees[5]). The magnitude of our estimate of the net $CO_2$–$N_2O$ effect of N-fixing trees at the global scale highlights the need for further research on the impact of N-fixing trees on soil $N_2O$ emissions.

Our analysis focused on a single global change factor—intensifying N deposition—due to its clear link to N supply. However, global change factors beyond N deposition such as increasing temperature, changing precipitation, and $CO_2$ fertilization could also influence the net $CO_2$–$N_2O$ effect of N-fixing trees. N-fixing trees are projected to increase in abundance due to increasing temperatures[36], which would amplify their net $CO_2$–$N_2O$ effect. Additionally, increasing temperatures will increase soil $N_2O$ emission rates[37,38]. N-fixing trees generally have a greater water use efficiency than non-fixing trees[39], and are more abundant in arid conditions[28,36,40], suggesting that changing precipitation could either increase or decrease N-fixing tree abundance and their net $CO_2$–$N_2O$ effect (although forecasted changes in precipitation in the United States and Mexico are projected to have only a minor influence on N-fixing tree abundance[36]). Additionally, soil moisture strongly controls soil $N_2O$ emission rates[37,38]. $CO_2$ fertilization has been suggested to promote N limitation via increased forest growth[41], although empirical evidence is mixed[42,43]. Intensifying N limitation could promote increasing N fixation rates[44,45] and a net $CO_2$–$N_2O$ cooling effect of N-fixing trees relative to non-fixing trees, although this

response could be limited by other nutrients[44,45]. Our study only addresses intensifying N deposition as it has a direct influence on N limitation, but other global change factors should also be considered for a comprehensive analysis of how N-fixing trees will mitigate or exacerbate climate change.

Forest management studies have recommended planting N-fixing trees during reforestation to alleviate regenerating forests from N limitation[46,47]. However, our study suggests that planting obligate and incomplete regulator N-fixing trees may actually exacerbate climate change relative to non-fixing trees under elevated N deposition rates. This finding complements recent empirical evidence that N-fixing trees might not promote forest growth[31–33]. However, we emphasize that in our study, the net $CO_2$–$N_2O$ effect of all forest ecosystems is a cooling effect (Supplementary Figure 3), and we are addressing the relative merit of N-fixing trees (with different N fixation strategies) vs. non-fixing trees. Furthermore, our analysis does not consider the merits of biodiversity or other site-specific factors that could influence the net $CO_2$–$N_2O$ effect of N-fixing trees.

Biological N fixation is a significant source of uncertainty in the climate projections of Earth System Models[48,49]. Our results suggest that including the regulation of biological N fixation in Earth System Models and explicitly considering soil $N_2O$ emissions, rather than $CO_2$ sequestration alone, could considerably decrease estimates of the extent to which global forests will mitigate climate change. Global forests currently sequester 2.4 Pg C $yr^{-1}$ (ref. [1]), representing a negative radiative forcing. Our analysis suggests that a single functional group, N-fixing trees, could decrease the magnitude of this negative radiative forcing of forests by up to 32% as N deposition intensifies. The theoretical modeling approach we employ here is only a basic framework for generating hypotheses and exploring their potential limits. We do not claim to have made accurate predictions for the net $CO_2$–$N_2O$ effect of N-fixing trees, but rather seek to stimulate discussion on their climate role and suggest further research. In particular, empirical work is necessary to quantify the net $CO_2$–$N_2O$ effect of N-fixing trees and improve its representation in Earth System Models, allowing the development of an accurate estimate of the extent to which N-fixing trees and global forests will mitigate or exacerbate climate change.

## Methods
**Model description.** Our model is an extension of a simple differential equation ecosystem model[18,50]. It includes a N-fixer biomass C pool ($B_F$, kg C $ha^{-1}$), a non-fixer biomass C pool ($B_0$, kg C $ha^{-1}$), a plant-unavailable soil N pool ($D$, kg N $ha^{-1}$; detritus), and a plant-available soil N pool ($A$, kg N $ha^{-1}$; nitrate, ammonium, and forms of organic N that are accessible to plants). The rates of change of these pools satisfy the following ordinary differential equations (represented by the box diagram in Supplementary Figure 5):

$$\frac{dB_F}{dt} = B_F\big(g_F(A, B_0, B_F) - \mu_F\big) \tag{1}$$

$$\frac{dB_0}{dt} = B_0\big(g_0(A, B_0, B_F) - \mu_0\big) \tag{2}$$

$$\frac{dD}{dt} = \frac{\mu_F}{\omega_F}B_F + \frac{\mu_0}{\omega_0}B_0 - (m + \varphi)D \tag{3}$$

$$\frac{dA}{dt} = I - kA + mD - \frac{B_F(g_F(A, B_0, B_F) - \omega_F F)}{\omega_F} - \frac{B_0 g_0(A, B_0, B_F)}{\omega_0} - \eta A \tag{4}$$

The per capita growth rates of $B_F$ and $B_0$ are represented by the functions $g_F$ and $g_0$, respectively:

$$g_F(A, B_0, B_F) = MIN\left[\omega_F(\nu_F A + F), \frac{\beta_F}{1 + \gamma_F(B_F + B_0)}\right] \tag{5}$$

$$g_0(A, B_0, B_F) = MIN\left[\omega_0 \nu_0 A, \frac{\beta_0}{1 + \gamma_0(B_F + B_0)}\right] \tag{6}$$

The growth rate of $B_i$ ($i = F$ represents N-fixers, $i = 0$ represents non-fixers) is determined by Liebig's law of the minimum[51]. When $B_i$ is N-limited, $g_i$ is a function of the nutrient use efficiency of N ($\omega_i$), N uptake rate ($\nu_i$), and, for $B_F$, N fixation rate per unit biomass C ($F$). When $B_i$ is not N-limited, $g_i$ is limited by some unspecified resource (such as phosphorus, light, or space), represented by a density-dependent function that decreases with increasing total biomass ($B_F + B_0$). For non-N-limited growth, $\beta_i$ is the maximum growth rate and $\gamma_i$ is the coefficient that determines the extent to which $g_i$ is decreased by total biomass. The parameter $\mu_i$ represents the turnover rate, $m$ represents the mineralization rate, $\varphi$ represents the loss rate of plant-unavailable soil N, $I$ represents the abiotic N input flux, $k$ represents the loss rate of plant-available soil N other than gaseous losses of $N_2O$ (leaching of all forms of plant-available soil N and gaseous losses of nitric oxide (NO), ammonium ($NH_3$), and nitrogen gas ($N_2$)), and $\eta$ represents the gaseous loss rate of plant-available soil N as $N_2O$. We assume that the $N_2O$ gaseous loss rate is a linear function of $A$, following 2006 IPCC Guidelines for National Greenhouse Gas Inventories[52]. Thus, the atmospheric $N_2O$ pool ($E$; in kg $N_2O$-N $ha^{-1}$) satisfies the following equation:

$$\frac{dE}{dt} = \eta A - \psi E \tag{7}$$

The parameter $\psi$ represents the atmospheric $N_2O$ removal rate (through photolysis and oxidation reactions[11]) and is the inverse of the lifetime of $N_2O$ in the atmosphere.

Different N fixation strategies (obligate, facultative, and incomplete regulator) are represented by the following equation, which gives N fixation rate per unit biomass C:

$$F = MAX\left[F_{min}, MIN\left[\frac{\beta_F}{\omega_F(1 + \gamma_F(B_F + B_0))} - \nu_F A, F_{max}\right]\right] \tag{8}$$

The parameter $F_{min}$ represents the sustained minimum N fixation rate, and thus describes the gradient of N fixation strategies from obligate N-fixers ($F_{min} = F_{max}$, i.e. $F$ is constant), to incomplete regulator N-fixers ($0 < F_{min} < F_{max}$), to facultative N-fixers ($F_{min} = 0$). The parameter $F_{max}$ is the maximum N fixation rate per unit biomass C.

**Model simulations.** Simulations of the model were conducted in R using the package deSolve. We parameterized our model for tropical, temperate, and boreal forests (Supplementary Table 2), and conducted the following simulations for each parameterization. We simulated four versions of the model (ecosystems containing only non-fixers, ecosystems containing non-fixers and obligate N-fixers, ecosystems containing non-fixers and facultative N-fixers, and ecosystems containing non-fixers and incomplete regulator N-fixers) for 100 years. We simulated each of the four versions of the model under three N deposition rates: past (low; pre-Anthropocene; from Galloway et al.[23]), recent (intermediate; 2001 and 2006; from Vet et al.[24] and Dentener et al.[12] respectively), and future N deposition rates (high; 2030 for the SRES A2 scenario[25]; from Dentener et al.[12]) (Supplementary Table 3). N deposition rates for tropical, temperate, and boreal forests were estimated using weighted averages with tropical, temperate, and boreal forest areas (from the 2015 Global Forest Resources Assessment[53]). The range of N deposition rates can also be representative of varying degrees of N enrichment from other sources (rock weathering N input, turnover, mineralization, etc.). Additionally, we simulated each of the four versions of the model under low, intermediate, and high initial soil N pool sizes (Supplementary Table 4).

**$CO_2$ effect, $N_2O$ effect, and net $CO_2$–$N_2O$ effect.** We calculated the $CO_2$ and $N_2O$ effects of the ecosystem with two complementary methods. The first method quantifies total change in the sizes of the biomass C pools and the atmospheric $N_2O$ pool, converting $N_2O$ to $CO_2$ radiative equivalents using global warming potentials. The second method quantifies net radiative forcing from continuous changes in the sizes of the biomass C pools and the atmospheric $N_2O$ pool. Both methods calculate the $CO_2$ and $N_2O$ effects of the ecosystem over 100 years, similar to the IPCC's SRES and Representative Concentration Pathways. The first method is easier to compare to studies of standing biomass C pools, whereas the second method gives a more accurate accounting of net radiative forcing. Results given in the main text are from the first method, but both methods give similar results. For the first method, the $CO_2$ and $N_2O$ effects of the ecosystem were calculated as follows:

$$CO_2\ \text{effect} = -\frac{((B_F(100) + B_0(100)) - (B_F(0) + B_0(0)))}{100\ yr} \times \frac{44\ kg\ CO_2}{12\ kg\ C} \tag{9}$$

$$N_2O\ \text{effect} = \frac{(E(100) - E(0))}{100\ yr} \times \frac{44\ kg\ N_2O}{28\ kg\ N} \times \frac{298\ kg\ CO_2}{kg\ N_2O} \tag{10}$$

The global warming potential of $N_2O$ over a 100 year time horizon[11] (298 kg $CO_2$ per kg $N_2O$) was used to find the $CO_2$ radiative equivalent of soil $N_2O$ emissions. The $CO_2$ effect and $N_2O$ effect are both given in units of kg $CO_2$ $ha^{-1}$ $yr^{-1}$.

For the second method, we adapted an equation for the radiative forcing of a continuous emission pulse from Alvarez et al.[54]:

$$CO_2 \text{ effect} = -\int_{t_E=0}^{t_E=100} RE_{CO_2} g_{CO_2}(t_E)\left((100-t_E)a_0 + \sum_{i=1}^{3} a_i \tau_{CO_2,i}\left(1 - e^{-\frac{100-t_E}{\tau_{CO_2,i}}}\right)\right)dt_E \tag{11}$$

$$N_2O \text{ effect} = \int_{t_E=0}^{t_E=100} RE_{N_2O} \frac{\eta A(t_E)}{\psi}\left(1 - e^{-\psi(100-t_E)}\right)dt_E \tag{12}$$

$g_{CO_2}(t_E)$ is the sequestration of $CO_2$ at time $t_E$. $a_i$ and $\tau_{CO_2,i}$ are constants and lifetimes respectively that represent the timescales of different $CO_2$ removal processes[55]. Removal of $CO_2$ by the terrestrial sink is already included in these $CO_2$ removal processes, and, as such, Eq. (11) is not an ideal representation of the $CO_2$ effect but is effective at demonstrating its general trend. $A(t_E)$ is the available soil N pool at time $t_E$. $RE_{GHG}$ is the radiative efficiency of the greenhouse gas and was calculated using the following formula from Myhre et al.[11] that converts radiative efficiency from units of W m$^{-2}$ ppbv$^{-1}$ (standard) to units of W m$^{-2}$ kg$^{-1}$:

$$RE_{GHG} = RE_{GHG,ppbv}\frac{M_A}{M_{GHG}}\frac{10^9}{T_M} \tag{13}$$

$RE_{GHG,ppbv}$ is the radiative efficiency in units of W m$^{-2}$ ppbv$^{-1}$, $M_A$ is the mean molar mass of air, $M_{GHG}$ is the molar mass of the greenhouse gas, and $T_M$ is the total mass of the atmosphere. Parameter values and descriptions are available in Supplementary Table 5. Results and figures corresponding to those available in the main text are displayed in Supplementary Table 6 and Supplementary Figure 6.

For both methods, the net $CO_2$–$N_2O$ effect reflects the balance of $CO_2$ sequestration and soil $N_2O$ emissions and is thus calculated as the sum of the $CO_2$ effect and $N_2O$ effect. A negative net $CO_2$–$N_2O$ effect indicates a cooling effect ($CO_2$ sequestration exceeds soil $N_2O$ emissions) and a positive net $CO_2$–$N_2O$ effect indicates a warming effect (soil $N_2O$ emissions exceed $CO_2$ sequestration).

**Model validity.** The model accurately estimates $CO_2$ sequestration and soil $N_2O$ emissions under recent N deposition rates. For tropical forests, the total biomass C equilibrium of the model is 124 Mg C ha$^{-1}$ (see Supplementary Note 1 for equilibria analysis), which is similar to Batterman et al.[3], which reported 128 Mg C ha$^{-1}$ in old growth tropical forests. For temperate forests, the total biomass C equilibrium of the model is 145 Mg C ha$^{-1}$, which is similar to Pregitzer et al.[56], which reported 171 Mg C ha$^{-1}$ in old growth temperate forests. For boreal forests, the total biomass C equilibrium of the model is 75 Mg C ha$^{-1}$, which is similar to Pregitzer et al.[56], which reported 81 Mg C ha$^{-1}$ in old growth boreal forests.

For tropical forests, the soil $N_2O$ emission rate of the model ranges between 0 and 6.97 kg $N_2O$-N ha$^{-1}$ yr$^{-1}$. This is less than the default value used by the IPCC for tropical forests[52] (16 kg $N_2O$-N ha$^{-1}$ yr$^{-1}$) but is similar to values from Stehfest and Bouwman[57] (1.37 kg $N_2O$-N ha$^{-1}$ yr$^{-1}$). For temperate forests, the soil $N_2O$ emission rate of the model ranges between 0 and 0.29 kg $N_2O$-N ha$^{-1}$ yr$^{-1}$. This is again less than the default value used by the IPCC for temperate forests[52] (8 kg $N_2O$-N ha$^{-1}$ yr$^{-1}$) but is similar to values from Stehfest and Bouwman[57] (0.64 kg $N_2O$-N ha$^{-1}$ yr$^{-1}$). For boreal forests, the soil $N_2O$ emission rate of the model ranges between 0 and 0.13 kg $N_2O$-N ha$^{-1}$ yr$^{-1}$. This is similar to values from Pihlatie et al.[58] (−0.67 to 0.39 kg $N_2O$-N ha$^{-1}$ yr$^{-1}$).

For tropical forests, the N fixation rate of the model ranges between 0 and 29 kg N ha$^{-1}$ yr$^{-1}$, which is similar to values from Batterman et al.[3] (0–29 kg N ha$^{-1}$ yr$^{-1}$), Sullivan et al.[20] (1.2–14.4 kg N ha$^{-1}$ yr$^{-1}$), and Winbourne et al.[59] (0.3–22.75 kg N ha$^{-1}$ yr$^{-1}$). For temperate forests, the N fixation rate of the model ranges between 0 and 10 kg N ha$^{-1}$ yr$^{-1}$, which is similar to values from Menge and Hedin[22] (0–11 kg N ha$^{-1}$ yr$^{-1}$). For boreal forests, the N fixation rate of the model ranges between 0 and 6 kg N ha$^{-1}$ yr$^{-1}$, which is similar to values from Blundon and Dale[60] (0.3 kg N ha$^{-1}$ yr$^{-1}$). Other reported N fixation rates for temperate forests[21,61] (33–150 kg N ha$^{-1}$ yr$^{-1}$) and boreal forests[62,63] (38–107 kg N ha$^{-1}$ yr$^{-1}$) are substantially higher, but N-fixing trees are often rare or absent in temperate and boreal forests[27]. As such, the average N fixation rates across temperate and boreal forests are likely within the range of the N fixation rates of our model.

**Global scale estimate.** We applied the net $CO_2$–$N_2O$ effect calculated with tropical, temperate, and boreal forest parameterizations to tropical, temperate, and boreal forest areas (from the 2015 Global Forest Resources Assessment[53]) respectively. Many forests are recovering from a past disturbance, imparting a variegated age distribution on global forests[64]. Because the net $CO_2$–$N_2O$ effect (Eqs. (9) and (10)) is averaged over the first 100 years of ecosystem succession, it roughly encompasses the age distribution of global forests.

**Reporting summary.** Further information on experimental design is available in the Nature Research Reporting Summary linked to this article.

## Code availability

Code used for analyses and figures has been archived in a GitHub repository (http://github.com/siankg/Nfixation_CO2N2O, https://doi.org/10.5281/zenodo.2576173).

## Data availability

Data sharing is not applicable to this study as no data were generated or analyzed, aside from the simulated data created by the model and code.

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

## Acknowledgements

We thank P. Akana, R. Arkebauer, T. Bytnerowicz, A. Huddell, A. Petach, A. Quebbeman, and B. Taylor for discussions and comments on the manuscript. This study was supported by the National Oceanic and Atmospheric Administration's Climate Program Office's Atmospheric Chemistry, Carbon Cycle, and Climate program, award NA15OAR4310065, and by the National Science Foundation, award DEB-1457650. We acknowledge the support of the Natural Sciences and Engineering Research Council of Canada. Cette recherche a été financée par le Conseil de recherches en sciences naturelles et en génie du Canada.

## Author contributions

S.K.-G. and D.M. designed the study. S.K.-G. performed the analysis and wrote the initial draft. Both authors contributed to the writing.

## Additional information

**Competing interests:** The authors declare no competing interests.

