## [Peer Review File · Nature Communications]

Reviewers' comments:

Reviewer #1 (Remarks to the Author):

Review of NCOMMS-18-14429: Nitrogen-fixing trees will exacerbate climate change under elevated nitrogen deposition

This manuscript reports model results that demonstrate the tradeoffs between C fixation and N gas loss as a result of N fixation. It posits the novel concept that, under certain conditions, the N fixed by symbiotic N fixing plants will be emitted back to the atmosphere as N₂O, a greenhouse gas that is far more potent than CO₂. Typically, symbiotic N-fixing plants are assumed to provide the N needed to meet N demand in ecosystems, and so symbiotic N-fixers are thought to help improve CO₂ uptake through photosynthesis by reducing N limitation. However, if the model presented here is correct, symbiotic N-fixers could actually offset CO₂ uptake by increased N₂O emissions.

I am supportive of this well written and well conceived manuscript, but it needs some refinement before it can be published. First, the title of the manuscript slightly oversells the negative impact of N-fixing plants on climate. The manuscript results are clear that under some combinations of N deposition (high) and N-fixation (obligate), N-fixers could represent a climate “problem.” The title leaves out some important nuance to the results. I understand the need to write brief and punchy titles, but I would encourage the authors to add “non-facultative nitrogen fixing trees” or “some nitrogen fixing trees” or something like that to make it clear that this problem is dependent on types of N fixers. Another way to spin this (and the authors have probably already considered it) is to state that N loss from N-fixers under high N deposition “offsets” C uptake rather than “exacerbates” climate change. That’s the universal result from the high N-deposition rate panel of Figure 1, across all N-fixing plants.

I understood the takeaways from Figure 2, but I found that the figure did not match the text between lines 82-89 very well. For example, the authors write (on line 84), “Under low N deposition, the sustained N fixation of obligate and incomplete regulator N-fixing trees facilitates non-fixing trees in meeting their N demand, amplifying ecosystem-scale CO₂ sequestration... (Fig 2b).” However, Figure 2b only shows the relationship between N fixation rate and CO₂ sequestration under low and high N deposition rates, rather than for different N fixation strategies. A similar issue arose in Fig. 2c, where it wasn’t clear what was meant by “after N limitation is alleviated.” This could be simply shown by an arrow pointing to the inflection point on the dashed line in the low N deposition rate that represents N limitation alleviation. They then go on to say this effect was most pronounced for obligate fixers, but Fig 2c, like Fig. 2b, does not show different N fixation strategies. I

encourage the authors to revise the figures to best reflect the take-home messages in the text, even if that means making the figures a little more complicated.

A theoretical model like this one (no matter how well it represents “reality”) generates hypotheses and tests compelling ideas. It is not, however, a truly prognostic tool. I encourage the authors to use this framework more explicitly. The pattern reported in this manuscript needs to be empirically tested and validated (and I would guess that the authors are working on this, given their upcoming ESA talk). But until that point, this model is the best we have, and its value lies in overturning paradigms. The changes required to reflect this difference would be subtle. For example, the authors use “demonstrate” in line 10 of the abstract, but this doesn’t really demonstrate it – the model posits, or suggests, that this effect could occur. Another place to consider adjusting text would be lines 15-16 of the abstract: instead of “as nitrogen deposition intensifies, nitrogen-fixing trees stimulate substantial soil N₂O emissions,” just include “as nitrogen deposition intensifies, our model hypothesizes that nitrogen-fixing trees will stimulate substantial soil N₂O emissions.” The model may be right, it may be right for the wrong reasons, or it might not be exactly right... but the value of this model and this manuscript lies in turning the ideas of N fixing trees upside down. Let it stand on that strength and avoid making it appear as though the model truly is reality. This is especially important with N₂O, which is a notoriously challenging gas to estimate annual fluxes of in the field. As the authors note, the range of estimates for a biome are large and not well constrained. Until we have better numbers from the field, models will be the way forward for analyses like this, but we need to be careful before we put too much weight on either the field or model results just yet.

Finally, the text in the final two paragraphs of the manuscript might benefit from some changes. The authors, reasonably, assume simple end-member scenarios in which all trees on Earth are one of the three fixation strategies. This is wildly inaccurate, but they are left with few other options without better spatial estimates of N fixation strategies, as they acknowledge (line 113). I think a few adjustments would help this land better. For example, on line 111-112, instead of saying “we estimated the net CO₂-N₂O effect of N-fixing trees at the global scale,” say something slightly different like “we sought to put boundaries on the possible net CO₂-N₂O effect of N-fixing trees at the global scale.” They have text about boundaries on line 117, but I think the point needs to be made immediately. As bounds, the “incomplete regulator” trees are not very helpful – they come in as a midpoint that quickly gets lost by the focus on the outer boundaries. Therefore, another important change might be to make a 3D response surface to this problem, showing the global net CO₂-N₂O effect (Y axis) given a fraction of obligate-facultative fixers (from 0-100% of each (X axis) and a range of N deposition rates (Z-axis). By cutting the incomplete regulators, the boundary conditions stay the same and the truth might lie somewhere in the middle of the 3D response surface. The authors could always include text that states what the net CO₂-N₂O effect would be if all trees were incomplete regulators (from Table 1). But the goal here should be on establishing the scale of the problem, rather than providing a single “scary” number (0.68 Pg CO₂e y⁻¹, or 28% of forest CO₂ sequestration) that will get too much traction. I encourage the authors to consider how this kind of a number would be received by a less informed audience of managers and policy makers. It could encourage people to harvest legumes out of forests that are not experiencing high N

deposition yet, simply to avoid this problem. I work on land management projects for climate mitigation, and I have seen first-hand how a single number like that can be misconstrued and guide well intentioned but misguided policy. A nice 3D figure might reduce the focus on the 28% offset of C sequestration and demonstrate that under many future scenarios, legumes do not endlessly fuel C sequestration in forests.

Minor comments:

The authors might benefit from citing the recently accepted work by Soper et al. (2018) in *Ecology* demonstrating that N₂O emissions are higher under high-foliar N trees than low-foliar N trees. Those authors do not explicitly tie this pattern to N fixers, but N fixers often have higher foliar N concentrations than non fixers. Soper et al. could provide limited empirical evidence for the model results. <https://esajournals.onlinelibrary.wiley.com/doi/epdf/10.1002/ecy.2434>

The Line 87: please replace “sustained” with “increasing.” As it is depicted in Fig. 2, it appears that the effect is due to increasing N fixation (moving right on the X-axis) rather than sustained N fixation, which continued N fixation or the result of an obligate strategy, perhaps.

The paragraph starting line 94 needs some minor editorial help. First, I recommend writing “...induces facultative N-fixing trees...” on lines 94-95, and citing Fig. 2a after “extent possible.” Then delete the parenthetical scenarios on lines 95-96, as it doesn’t make sense to talk about the effect of increased N supply to ecosystems as a regulator of fixation if fixation is not regulated by N supply.

On line 97-98, “...under high N deposition rates, N fixing trees have a lower abundance than under low N deposition rates.” Is this only true for facultative plants? I looked closely at Extended Data Fig 2 and it appears that is the case. Obligate legume abundance seemed to be little changed among N deposition scenarios.

Figure 1 didn’t print well in black and white – the bars were almost invisible. Try a different color/shading scheme.

I hope that these suggestions are useful to the authors and I commend them on a job well done with this manuscript. On the whole, it represents a well developed test of the hypothesis that N-fixing plants only improve climate change scenarios in the future. I think this manuscript will be a valuable contribution to the N-fixation literature.

Reviewer #2 (Remarks to the Author):

The paper investigates the influence of biological nitrogen fixation (BNF) on the net carbon sequestration rate in forests (= C sequestration – N₂O emission). The authors argue that BNF can both increase and decrease carbon sequestration, depending on the type of BNF and rate of atmospheric N deposition. To test this the authors use a theoretical mathematical model and scale up to the globe. Their final comment is that BNF needs to be included when modelling the forest sink strength of CO₂.

I do agree with the final comment (line 20-23), and I do agree that this paper is of interest to the wider scientific audience. It provides an interesting theory which should stimulate a lot of thought and hopefully lead to more scientific interest in this important area of research.

My problem with this paper is the bold use of a theoretical model to scale up to the globe and suggest that by 2030 atmospheric N deposition together with BNF will reduce CO₂ sequestration in forests by 28%. This number and the values in Table 1 and Figure 1 & 2 are highly uncertain. To me this uncertainty does not come across, and the 28% statement should either be removed or come with a clearly stated 'health warning'.

I would like to see validation of this models against data from tropical, temperate and arctic forests. Simple things, like C sequestration rates, N₂O emission rates, denitrification ratios of N₂O/N₂, BNF rates and the split between climate zones, forest types etc. Ok, there are only few data available, but some validation would certainly boost my confidence in the value of this paper.

Generally this paper is well written. Some of the references and data should be updated, i.e. is there not a more recent global N dep map than the Dentener et al 2006? What is a Jacobian system? The mathematical equations in the supplement do not include the necessary information to identify the Roman and Greek letters. I know this info is in the main text, but should also appear in the supplement.

In conclusion, although the idea is very interesting, I do not support the publication of this paper, unless the statements can be validated by data or by calibrated process based models.

Reviewer #1 (Remarks to the Author):

Review of NCOMMS-18-14429: Nitrogen-fixing trees will exacerbate climate change under elevated nitrogen deposition

This manuscript reports model results that demonstrate the tradeoffs between C fixation and N gas loss as a result of N fixation. It posits the novel concept that, under certain conditions, the N fixed by symbiotic N fixing plants will be emitted back to the atmosphere as N₂O, a greenhouse gas that is far more potent than CO₂. Typically, symbiotic N-fixing plants are assumed to provide the N needed to meet N demand in ecosystems, and so symbiotic N-fixers are thought to help improve CO₂ uptake through photosynthesis by reducing N limitation. However, if the model presented here is correct, symbiotic N-fixers could actually offset CO₂ uptake by increased N₂O emissions.

I am supportive of this well written and well conceived manuscript, but it needs some refinement before it can be published.

AUTHORS' RESPONSE: We thank the reviewer for this comment and address the individual points below.

First, the title of the manuscript slightly oversells the negative impact of N-fixing plants on climate. The manuscript results are clear that under some combinations of N deposition (high) and N-fixation (obligate), N-fixers could represent a climate “problem.” The title leaves out some important nuance to the results. I understand the need to write brief and punchy titles, but I would encourage the authors to add “non-facultative nitrogen fixing trees” or “some nitrogen fixing trees” or something like that to make it clear that this problem is dependent on types of N fixers.

AUTHORS' RESPONSE: This is a great point, and we thank the reviewer for this suggestion. We have considered a number of possible modifications to the title, with the goals of (1) Conveying the main message, (2) not over-selling the results, and (3) keeping the title compelling enough to draw in readers. We hope you agree that our new title, “Nitrogen-fixing trees could exacerbate climate change under elevated nitrogen deposition,” balances these goals. The main change is the word “could” instead of “will,” which we believe conveys the conditional nature of our results without getting too far into the weeds.

Another way to spin this (and the authors have probably already considered it) is to state that N loss from N-fixers under high N deposition “offsets” C uptake rather than “exacerbates” climate change. That’s the universal result from the high N-deposition rate panel of Figure 1, across all N-fixing plants.

AUTHORS' RESPONSE: We have considered a number of alternative wordings, many of which have merit. We opted for “exacerbates” and “mitigates climate change” for two

reasons. First, we are comparing the cooling and warming effects of CO₂ and N₂O. To us, “exacerbates climate change” keeps the focus on these cooling and warming effects, rather than on the carbon cycle. Second, we are discussing the influence of N-fixing trees relative to non-fixing trees, rather than the absolute influence of N-fixing trees on climate change. For N-fixing trees that exacerbate climate change relative to non-fixing trees, soil N₂O emissions do not necessarily offset CO₂ sequestration. Rather the offset of CO₂ sequestration by soil N₂O emissions for N-fixing trees is greater than the offset of CO₂ sequestration by soil N₂O emissions for non-fixing trees, i.e. N-fixing trees “exacerbate climate change” more than non-fixing trees but do not necessarily “exacerbate climate change” overall. We believe that “mitigate” and “exacerbate climate change” would be less prone to this misinterpretation than “offset”. In this revision we have added clarification in the main text (lines 29-36 and 106-114), and added Extended Data Fig. 3 to clarify these points.

I understood the takeaways from Figure 2, but I found that the figure did not match the text between lines 82-89 very well. For example, the authors write (on line 84), “Under low N deposition, the sustained N fixation of obligate and incomplete regulator N-fixing trees facilitates non-fixing trees in meeting their N demand, amplifying ecosystem-scale CO₂ sequestration... (Fig 2b).” However, Figure 2b only shows the relationship between N fixation rate and CO₂ sequestration under low and high N deposition rates, rather than for different N fixation strategies. A similar issue arose in Fig. 2c, where it wasn’t clear what was meant by “after N limitation is alleviated.” This could be simply shown by an arrow pointing to the inflection point on the dashed line in the low N deposition rate that represents N limitation alleviation. They then go on to say this effect was most pronounced for obligate fixers, but Fig 2c, like Fig. 2b, does not show different N fixation strategies. I encourage the authors to revise the figures to best reflect the take-home messages in the text, even if that means making the figures a little more complicated.

AUTHORS’ RESPONSE: Great idea. We have modified Fig. 2, which we hope clarifies this point. We have added vertical lines to Fig. 2b and 2c to indicate the average N fixation rate over 100 years for the three N fixation strategies examined in the model, as well as brackets to indicate the range of N fixation rates over 100 years for the three N fixation strategies examined in the model. We have added vertical lines at 0 to indicate non-fixers. We have also modified the caption of Fig. 2 to more clearly explain the results with respect to N limitation.

A theoretical model like this one (no matter how well it represents “reality”) generates hypotheses and tests compelling ideas. It is not, however, a truly prognostic tool. I encourage the authors to use this framework more explicitly. The pattern reported in this manuscript needs to be empirically tested and validated (and I would guess that the authors are working on this, given their upcoming ESA talk). But until that point, this model is the best we have, and its value lies in overturning paradigms. The changes required to reflect this difference would be subtle. For example, the authors use “demonstrate” in line 10 of the abstract, but this doesn’t really demonstrate it – the model posits, or suggests, that this effect could occur. Another place to consider adjusting text would be lines 15-16 of the abstract: instead of “as nitrogen deposition intensifies, nitrogen-fixing trees stimulate substantial soil N₂O emissions,” just include “as nitrogen deposition intensifies, our model hypothesizes that nitrogen-fixing trees will stimulate substantial soil N₂O emissions.” The model may be right, it may be right for the wrong reasons,

or it might not be exactly right... but the value of this model and this manuscript lies in turning the ideas of N fixing trees upside down. Let it stand on that strength and avoid making it appear as though the model truly is reality. This is especially important with N₂O, which is a notoriously challenging gas to estimate annual fluxes of in the field. As the authors note, the range of estimates for a biome are large and not well constrained. Until we have better numbers from the field, models will be the way forward for analyses like this, but we need to be careful before we put too much weight on either the field or model results just yet.

AUTHORS' RESPONSE: Thank you for this suggestion. We agree, and we have now edited the text throughout to emphasize that the goal of our analysis is not to make accurate predictions but to generate and explore hypotheses of how N-fixing trees could mitigate or exacerbate climate change. We have changed wording in the lines indicated and throughout the text to more clearly indicate the role of a theoretical model as opposed to data and the limitations of our analysis.

Finally, the text in the final two paragraphs of the manuscript might benefit from some changes. The authors, reasonably, assume simple end-member scenarios in which all trees on Earth are one of the three fixation strategies. This is wildly inaccurate, but they are left with few other options without better spatial estimates of N fixation strategies, as they acknowledge (line 113). I think a few adjustments would help this land better. For example, on line 111-112, instead of saying “we estimated the net CO₂-N₂O effect of N-fixing trees at the global scale,” say something slightly different like “we sought to put boundaries on the possible net CO₂-N₂O effect of N-fixing trees at the global scale.” They have text about boundaries on line 117, but I think the point needs to be made immediately. As bounds, the “incomplete regulator” trees are not very helpful – they come in as a midpoint that quickly gets lost by the focus on the outer boundaries. Therefore, another important change might be to make a 3D response surface to this problem, showing the global net CO₂-N₂O effect (Y axis) given a fraction of obligate-facultative fixers (from 0-100% of each (X axis) and a range of N deposition rates (Z-axis). By cutting the incomplete regulators, the boundary conditions stay the same and the truth might lie somewhere in the middle of the 3D response surface. The authors could always include text that states what the net CO₂-N₂O effect would be if all trees were incomplete regulators (from Table 1). But the goal here should be on establishing the scale of the problem, rather than providing a single “scary” number (0.68 Pg CO₂e y⁻¹, or 28% of forest CO₂ sequestration) that will get too much traction. I encourage the authors to consider how this kind of a number would be received by a less informed audience of managers and policy makers. It could encourage people to harvest legumes out of forests that are not experiencing high N deposition yet, simply to avoid this problem. I work on land management projects for climate mitigation, and I have seen first-hand how a single number like that can be misconstrued and guide well intentioned but misguided policy. A nice 3D figure might reduce the focus on the 28% offset of C sequestration and demonstrate that under many future scenarios, legumes do not endlessly fuel C sequestration in forests.

AUTHORS' RESPONSE: This is an excellent suggestion. We have added a figure (Fig. 3) as suggested, which includes both a 3D panel and a 2D panel (for those readers who, like the senior author on the paper, find 3D figures challenging to interpret) with cross sections of the extremes of the surface in the 3D panel. Additionally, we have clarified that the maximum and

minimum of the three scenarios provide boundaries to a global scale estimate of the net CO₂-N₂O effect of N-fixing trees relative to non-fixing trees earlier in the text. We have also added cautionary words to the Discussion on the interpretation of these results for forest management.

Minor comments:

The authors might benefit from citing the recently accepted work by Soper et al. (2018) in Ecology demonstrating that N₂O emissions are higher under high-foliar N trees than low-foliar N trees. Those authors do not explicitly tie this pattern to N fixers, but N fixers often have higher foliar N concentrations than non fixers. Soper et al. could provide limited empirical evidence for the model results. <https://esajournals.onlinelibrary.wiley.com/doi/epdf/10.1002/ecy.2434>

AUTHORS' RESPONSE: Thank you for pointing us to this interesting new study. We had not seen it before submitting the initial version of our manuscript, but find it to be an excellent and relevant study. We have now cited it in the "Introduction" and "Discussion" sections.

The Line 87: please replace "sustained" with "increasing." As it is depicted in Fig. 2, it appears that the effect is due to increasing N fixation (moving right on the X-axis) rather than sustained N fixation, which continued N fixation or the result of an obligate strategy, perhaps.

AUTHORS' RESPONSE: We agree that Fig. 2 and its explanation were unclear. We hope that we have addressed this in the new version of Fig. 2. Obligate and incomplete regulator N-fixing trees sustain N fixation after satisfying their N demand. This contributes to a high average N fixation rate over 100 years which is indicated by the vertical lines in Fig. 2b and 2c.

The paragraph starting line 94 needs some minor editorial help. First, I recommend writing "...induces facultative N-fixing trees..." on lines 94-95, and citing Fig. 2a after "extent possible." Then delete the parenthetical scenarios on lines 95-96, as it doesn't make sense to talk about the effect of increased N supply to ecosystems as a regulator of fixation if fixation is not regulated by N supply.

AUTHORS' RESPONSE: We have changed the wording of this sentence and cited Fig. 2a earlier. We discuss obligate and incomplete regulator N-fixers because, even though they do not completely down-regulate N fixation, the interaction between N fixation strategy and increased N supply via elevated N deposition leads to interesting differences between the net CO₂-N₂O effects of ecosystems with different N fixation strategies (see Fig. 1, High N deposition rate panel).

On line 97-98, "...under high N deposition rates, N fixing trees have a lower abundance than under low N deposition rates." Is this only true for facultative plants? I looked closely at Extended Data Fig. 2 and it appears that is the case. Obligate legume abundance seemed to be little changed among N deposition scenarios.

AUTHORS' RESPONSE: Thank you for catching this. Only facultative and incomplete regulator N-fixing trees have a higher maximum abundance under low N deposition rates

than under high N deposition rates.

Figure 1 didn't print well in black and white – the bars were almost invisible. Try a different color/shading scheme.

AUTHORS' RESPONSE: Thank you for identifying this. We have changed the colour scheme, which we hope is now more suitable for black and white.

I hope that these suggestions are useful to the authors and I commend them on a job well done with this manuscript. On the whole, it represents a well developed test of the hypothesis that N-fixing plants only improve climate change scenarios in the future. I think this manuscript will be a valuable contribution to the N-fixation literature.

AUTHORS' RESPONSE: Thank you!

Reviewer #2 (Remarks to the Author):

The paper investigates the influence of biological nitrogen fixation (BNF) on the net carbon sequestration rate in forests (= C sequestration – N₂O emission). The authors argue that BNF can both increase and decrease carbon sequestration, depending on the type of BNF and rate of atmospheric N deposition. To test this the authors use a theoretical mathematical model and scale up to the globe. Their final comment is that BNF needs to be included when modelling the forest sink strength of CO₂.

I do agree with the final comment (line 20-23), and I do agree that this paper is of interest to the wider scientific audience. It provides an interesting theory which should stimulate a lot of thought and hopefully lead to more scientific interest in this important area of research.

AUTHORS' RESPONSE: We thank the reviewer for this comment.

My problem with this paper is the bold use of a theoretical model to scale up to the globe and suggest that by 2030 atmospheric N deposition together with BNF will reduce CO₂ sequestration in forests by 28%. This number and the values in Table 1 and Figure 1 & 2 are highly uncertain. To me this uncertainty does not come across, and the 28% statement should either be removed or come with a clearly stated 'health warning'.

AUTHORS' RESPONSE: Thank you for this suggestion. We agree, and have modified the text accordingly. Reviewer #1 had a similar comment; please see our response above for a more detailed explanation of the changes we have made.

I would like to see validation of this models against data from tropical, temperate and arctic forests. Simple things, like C sequestration rates, N₂O emission rates, denitrification ratios of N₂O/N₂, BNF rates and the split between climate zones, forest types etc. Ok, there are only few data available, but some validation would certainly boost my confidence in the value of this paper.

AUTHORS' RESPONSE: We agree that validation is important. In our original submission we had included validation of C sequestration rates and N₂O emission rates in the Supplementary Information. We agree that this information belongs in the main text and, in our revision, have included it in the "Methods" section (lines 362-378). Additionally, in our revision we added validation of BNF rates of the model against data from tropical, temperate and boreal forests. We did not validate denitrification ratios of N₂O/N₂ because they are not relevant to our results. Our model includes N₂O emissions, but does not model them explicitly as a fraction of total denitrification. The model accurately estimates C sequestration rates, N₂O emission rates and BNF rates.

Generally this paper is well written. Some of the references and data should be updated, i.e is there not a more recent global N dep map than the Dentener et al 2006?

AUTHORS' RESPONSE: We find Dentener et al. 2006 to be the best published comparison of current N deposition rates to future N deposition rates at the global scale. Although Vet et al. 2014 provides a more recent estimate of current N deposition rates at the global scale, it does not make predictions for future N deposition rates. We prefer to use estimates from Dentener et al. 2006 to maintain consistency between current N deposition rates and future N deposition rates. However, we redid our analysis with the current N deposition rates from Vet et al. 2014 and have included it in the Supplementary Information. Additionally, we have added a new figure that displays the global net CO₂-N₂O effect of N-fixing trees relative to non-fixing trees over a range of N deposition rates. We used the lesser of the two estimates of current N deposition rate (between Dentener et al. 2006 and Vet et al. 2014) as the minimum N deposition rate displayed in the figure. As such, the global net CO₂-N₂O effect of N-fixing trees relative to non-fixing trees at both the Dentener et al. 2006 and Vet et al. 2014 estimates of current N deposition rate are displayed in Fig. 3.

What is a Jacobian system? The mathematical equations in the supplement do not include the necessary information to identify the Roman and Greek letters. I know this info is in the main text, but should also appear in the supplement.

AUTHORS' RESPONSE: Thank you for identifying this. For greater accessibility, we briefly explained the use of a Jacobian in analyzing systems of differential equations, and we have also included a reference for readers that are interested in learning more. To explain the Roman and Greek letters, we referenced the "Methods" section and Extended Data Table 1 which describes each of the parameters.

In conclusion, although the idea is very interesting, I do not support the publication of this paper, unless the statements can be validated by data or by calibrated process based models.

REVIEWERS' COMMENTS:

Reviewer #1 (Remarks to the Author):

I reviewed this manuscript previously (Reviewer 1) and I appreciate the many efforts made by the authors to revise the manuscript in accordance with the two reviews. I think this manuscript is much stronger as a result of these changes and I encourage its acceptance pending some minor revisions.

The new text addressing reviewer concerns on lines 29-36 is awkward and a little unclear. I think it might be better placed in the discussion, after the results have been presented. For example, the comment that "we are not suggesting that N-fixing trees can or will modify the cooling direction of the net CO₂-N₂O effect of forests" would be better used after those results are shown based on Extended Data Figure 3. The question introduced here (how N-fixing trees modify CO₂ sequestration in comparison to how they modify soil N₂O emissions relative to non-fixing trees) should be cut from the introduction. The idea here should be incorporated into the questions posed on lines 27-29 and line 58-60.

Line 37: I recommend editing this sentence to say "...offer insight into the possible net CO₂-N₂O effect of N enrichment."

Line 57: Please consider editing this sentence to "...calls for the explicit consideration of N fixation strategies when estimating the net CO₂-N₂O effect of forests."

Lines 121-122 and Fig 2b/c: Please make it more clear that these vertical lines in the figures are associated with different N fixation strategies. I got that from the text, but not from the figure or the caption.

Line 140: "Which increase indefinitely" – explain more. N₂O emissions increase with increasing what?

Line 152: "...in Global forests...?" I understand but it might sound better to say "in forests around the globe" or something like that.

Line 179: Consider "However, the magnitude of this estimate suggests that ..."

Line 211: "...studies of the extent to which and the controls under which..." while strictly grammatically correct is awkward. Perhaps try rephrasing.

Line 214: Delete "order of"

Line 237-239: There are too many phrases in this sentence. Break it up, perhaps?

Figure 1 (and similar figures): Please consider adding labels in the negative and positive areas of the graphs that say "Cooling" and "Warming", respectively. This is in the figure caption but the figure might benefit from having it included visually too.

Figure 3: Thank you for including the 3D graph, as well as taking my other suggestions seriously. I appreciate the revisions made here. Congratulations on a compelling and thought-provoking contribution to the literature.

Reviewer #2 (Remarks to the Author):

The authors have done a god job addressing the reviewer comments. Good that they have toned down their claims regarding the 28% reduction. Thank you!

I have 2 minor points I would like you to deal with:

Table 1: 3rd row & 3rd column: add 0 into this cell

Line 632: Vet et al. 2014 or Dentener et al. 2006. Adding both references is a bit confusing.

Presumably both provide the same result? If so replace 'or' with 'and'

So, overall I am very happy with this paper, and would recommend it for publication

Reviewer #1 (Remarks to the Author):

I reviewed this manuscript previously (Reviewer 1) and I appreciate the many efforts made by the authors to revise the manuscript in accordance with the two reviews. I think this manuscript is much stronger as a result of these changes and I encourage its acceptance pending some minor revisions.

The new text addressing reviewer concerns on lines 29-36 is awkward and a little unclear. I think it might be better placed in the discussion, after the results have been presented. For example, the comment that “we are not suggesting that N-fixing trees can or will modify the cooling direction of the net CO₂-N₂O effect of forests” would be better used after those results are shown based on Extended Data Figure 3. The question introduced here (how N-fixing trees modify CO₂ sequestration in comparison to how they modify soil N₂O emissions relative to non-fixing trees) should be cut from the introduction. The idea here should be incorporated into the questions posited on lines 27-29 and line 58-60.

AUTHORS' RESPONSE: We agree that this concept should be introduced later in the text and we have moved these sentences to the final paragraph of the introduction. We combined the final question of lines 29-36 in the previous draft with the questions posited in line 58-60 in the previous draft (lines 69-71 in the new draft). We chose to keep this information in the introduction instead of in the results or discussion because we wanted to emphasize early in the manuscript that we are not suggesting that N-fixing trees could cause forests to have a net warming effect, which was unclear in the previous draft.

Line 37: I recommend editing this sentence to say “...offer insight into the possible net CO₂-N₂O effect of N enrichment.”

AUTHORS' RESPONSE: We understand your point here – there is still quite a bit of uncertainty about the net CO₂-N₂O effect of N enrichment. However, we feel that “offer insight into ...” already implies a lack of uncertainty about the true nature of the net CO₂-N₂O effect of N enrichment, so we have retained our original wording.

Line 57: Please consider editing this sentence to “...calls for the explicit consideration of N fixation strategies when estimating the net CO₂-N₂O effect of forests.

AUTHORS' RESPONSE: We agree that this wording is better and we have changed the wording of this sentence.

Lines 121-122 and Fig 2b/c: Please make it more clear that these vertical lines in the figures are associated with different N fixation strategies. I got that from the text, but not from the figure or the caption.

AUTHORS' RESPONSE: Thank you for pointing out that this was unclear. We have changed the wording of the legend of Figure 2 to clearly describe the vertical lines and corresponding brackets.

Line 140: “Which increase indefinitely” – explain more. N₂O emissions increase with increasing what?

AUTHORS' RESPONSE: Thank you for pointing out that this was unclear. We have clarified the sentence; it now reads “which increase indefinitely with increasing N fixation”.

Line 152: “...in Global forests...?” I understand but it might sound better to say “in forests around the globe” or something like that.

AUTHORS' RESPONSE: We have changed the wording of this sentence to “forests around the globe include an assemblage of...”.

Line 179: Consider “However, the magnitude of this estimate suggests that ...”

AUTHORS' RESPONSE: We have changed the wording of this sentence as the reviewer suggests.

Line 211: “...studies of the extent to which and the controls under which...” while strictly grammatically correct is awkward. Perhaps try rephrasing.

AUTHORS' RESPONSE: We agree that this was awkward and have cut “and the controls under which”.

Line 214: Delete “order of”

AUTHORS' RESPONSE: We have changed the wording of this sentence as the reviewer suggests.

Line 237-239: There are too many phrases in this sentence. Break it up, perhaps?

AUTHORS' RESPONSE: We broken up the sentence, as suggested.

Figure 1 (and similar figures): Please consider adding labels in the negative and positive areas of the graphs that say “Cooling” and “Warming”, respectively. This is in the figure caption but the figure might benefit from having it included visually too.

AUTHORS' RESPONSE: We chose not to include the words cooling and warming on the axes of the figures because we wanted to clarify alongside the description of “cooling effects” and “warming effects” that these effects are relative to non-fixing trees given that this concept was unclear in the previous draft. We hope that the colours “red” and “blue” suggest “warming” and “cooling” respectively.

Figure 3: Thank you for including the 3D graph, as well as taking my other suggestions seriously. I appreciate the revisions made here. Congratulations on a compelling and thought-provoking contribution to the literature.

AUTHORS' RESPONSE: Thank you for these comments and your previous comments on our manuscript!

Reviewer #2 (Remarks to the Author):

The authors have done a god job addressing the reviewer comments. Good that they have toned down their claims regarding the 28% reduction. Thank you!

I have 2 minor points I would like you to deal with:
Table 1: 3rd row & 3rd column: add 0 into this cell

AUTHORS' RESPONSE: We have added a "NA" to demonstrate that a value for the global net CO₂-N₂O effect of N-fixing trees is not applicable to a global forest composition of non-fixers.

Line 632: Vet et al. 2014 or Dentener et al. 2006. Adding both references is a bit confusing. Presumably both provide the same result? If so replace 'or' with 'and'

AUTHORS' RESPONSE: Nitrogen deposition rates given in both Vet et al. 2014 and Dentener et al. 2006 were used to generate Figure 3. For boreal forests we used the N deposition rate given in Dentener et al. 2006 (which was lower than the N deposition rate for boreal forests given in Vet et al. 2014), and for temperate and tropical forests we used the N deposition rates given in Vet et al. 2014 (which were lower than the N deposition rates for temperate and tropical forests respectively given in Dentener et al. 2006). As such we displayed the full range of recent N deposition rate estimates in Figure 3.

So, overall I am very happy with this paper, and would recommend it for publication

AUTHORS' RESPONSE: Thank you for these comments and your previous comments on our manuscript!